# Not just trash birds: Quantifying avian diversity at landfills using community science data

Zachary J. Arnold[1]*, Seth J. Wenger[1,2], Richard J. Hall[1,3]

1 Odum School of Ecology, University of Georgia, Athens, GA, United States of America, 2 The River Basin Center, University of Georgia, Athens, GA, United States of America, 3 Department of Infectious Diseases, College of Veterinary Medicine, University of Georgia, Athens, GA, United States of America

* zja83037@uga.edu

**Data Availability Statement:** The data underlying this study are available from the eBird database following the protocol outlined in the Methods section.

## Abstract

Landfills provide seasonally reliable food resources to many bird species, including those perceived to be pest or invasive species. However, landfills often contain multiple habitat types that could attract diverse species, including those of conservation concern. To date, little is known about the characteristics and composition of bird communities at landfills relative to local and regional pools. Here we used the community science database eBird to extract avian species occurrence data at landfills across the US. We compared species richness and community similarity across space in comparison to similarly-sampled reference sites, and further quantified taxonomic and dietary traits of bird communities at landfills. While landfills harbored marginally lower species richness than reference sites (respective medians of 144 vs 160), landfill community composition, and its turnover across space, were similar to reference sites. Consistent with active waste disposal areas attracting birds, species feeding at higher trophic levels, especially gulls, were more frequently observed at landfills than reference sites. However, habitat specialists including two declining grassland species, Eastern Meadowlark (*Sturnella magna*) and Savannah Sparrow (*Passerculus sandwichensis*), as well as migratory waterfowl, were more frequently encountered at landfills than reference sites. Together, these results suggest that landfills harbor comparable avian diversity to neighboring sites, and that habitats contained within landfill sites can support species of conservation concern. As covered landfills are rarely developed or forested, management of wetlands and grasslands at these sites represents an opportunity for conservation.

## Introduction

Land use change associated with human activity has dramatically shaped global biodiversity, and has been implicated as a driver of large-scale declines in birds [1,2]. Some human-modified habitats, including farmland, urban parks and backyards, attract birds by providing food subsidies or by mimicking or preserving fragments of natural habitats, potentially mitigating these declines. However, human habitat modification can also homogenize ecological

**Funding:** The authors received no specific funding for this work.

**Competing interests:** The authors have declared that no competing interests exist.

communities over space, so that pairs of distant human-modified habitats have a higher proportion of shared species than pairs of intact habitats in similar locations [3] Human-modified habitats may support predominantly human-adapted generalist or invasive species, contributing to biotic homogenization of bird [4–6], plant [7], and insect [8] communities. Nevertheless, as human populations continue to grow and natural land cover declines, human-dominated landscapes are emerging as an important element in biodiversity conservation [9].

Landfills remain the commonest method for disposing of human waste, and attract a variety of birds [10]. The large quantities of food scraps found in active areas of trash disposal (henceforth active sites) at landfills can dramatically alter bird ecology, with consequences for their populations and health. These food resources at active sites support carnivorous and scavenging species including White Stork (*Ciconia ciconia*) [11], Bald Eagle, (*Haliaeetus leucocephalus*) [12], Ring-billed Gull (*Larus delawarensis*) [13], and Turkey Vulture (*Cathartes aura*) [14]. Additionally, landfill food subsidies can compensate for reductions in natural food availability and are thought to contribute to global increases in gull populations in spite of crashes in their natural marine-derived prey [15,16], and support imperiled species such as the endangered Greater Adjutant Stork (*Leptoptilos dubius*) [17]. Landfills may also negatively influence biodiversity, by reducing abundance of sensitive species, directly through exposure to contaminants and pathogens [10], and indirectly by supporting human-adapted and exotic invasive species that outcompete specialist, range-restricted, and native species [12,13].

Aside from active trash disposal sites, landfills often contain habitats that could support species which do not directly utilize refuse (Fig 1). After active sites are filled and covered with geosynthetic liners, clay, and soil, grasses are grown on top of the refuse and are mowed regularly to prevent the establishment of deep-rooting woody vegetation that is believed to damage

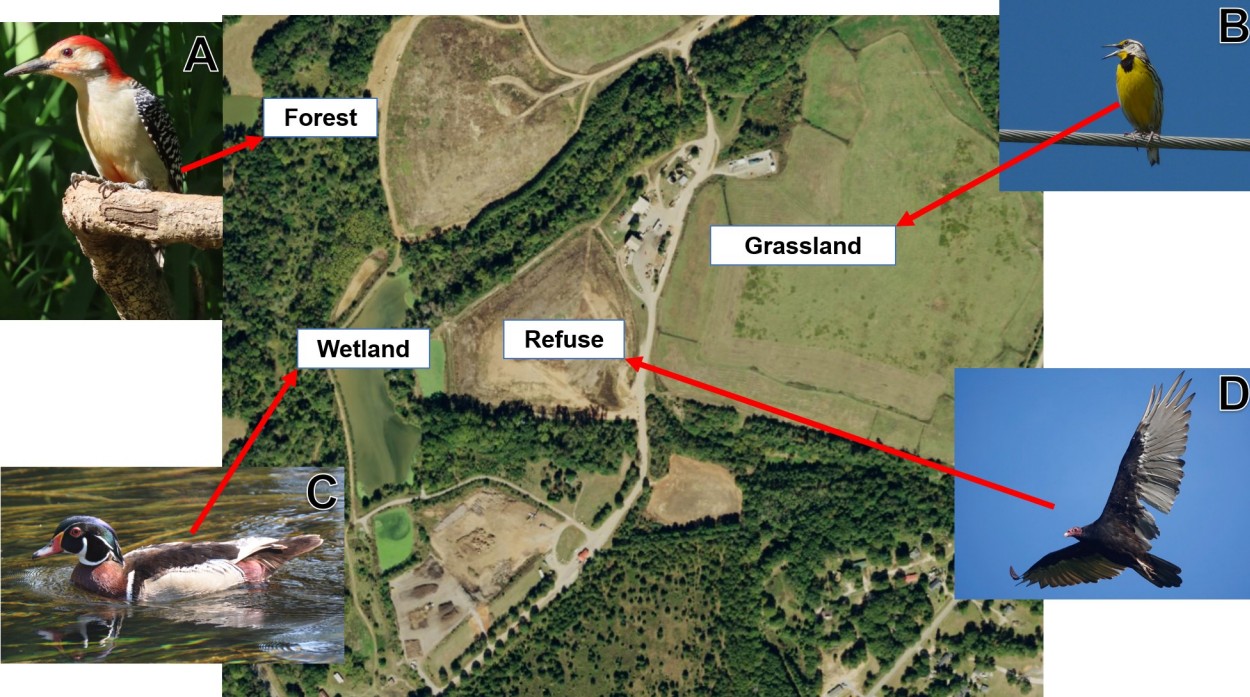

**Fig 1. The variety of habitats at landfills and the birds that use them.** National Agriculture Imagery Program (NAIP) imagery of Athens-Clarke Co. Landfill in Georgia, USA, depicting the various habitat types available at many landfills, including species that use each habitat. (A) Red-bellied woodpecker, *Melanerpes carolinus*; (B) Eastern meadowlark, *Sturnella magna*; (C) Wood duck, *Aix sponsa*; (D) Turkey vulture, *Cathartes aura*.

the capping system [18] As a result, these sites are maintained as grasslands for many years, providing potential habitat for declining specialist species [19]. Additionally, landfills provide constructed wetlands, built to capture run-off and protect nearby waterways [20] as well as remnants of the historical habitat on the periphery of landfill properties. Restrictions on subsequent land use provide potential opportunities for restoration and conservation, including parks, hiking trails, and nature preserves [21,22]. Studies that have looked at birds using active sites at landfills have primarily focused on cosmopolitan and those perceived as nuisance species [15–17,23,24], and thus the overall effects of landfills on bird community composition are not well understood. Quantifying bird communities at landfills will determine whether habitat specialists use additional habitat types beyond active sites, and thus is a crucial first step for determining landfill management practices that maintain bird diversity at both active landfills and their post-closure uses.

Birds are an excellent taxon to explore patterns of diversity, because they are widespread, visible, and increasingly, large numbers of birdwatchers use community science (also known as citizen science, henceforth referred to as CS) databases such as eBird [25] to document their observations. In particular, birdwatchers frequently visit landfills in an attempt to locate unusual species attracted to active landfills, such as out-of-range gulls and crows, as well as migratory waterfowl, shorebirds, and grassland birds that use the mosaic of habitat types present [26]. Here we describe patterns of taxonomic, habitat, and dietary diversity of birds at frequently visited landfills across the continental US, in comparison to nearby reference sites and county-level species pools. Reference sites were within the same county as each landfill and had been similarly sampled to allow for direct comparisons of average species richness and patterns of species turnover, in landfills versus "natural" sites. Comparison of landfills to county-level species pools enable us to determine whether species with certain dietary traits or habitat associations are more frequently encountered at landfills than expected relative to their occurrence in the county-level species pool.

We predicted that if landfills were dominated by widespread, generalist, human-adapted species, we would see: (i) lower species richness at landfills than at reference sites; (ii) higher species similarity among pairs of landfills compared with pairs of reference sites; and (iii) more records of carnivore and scavenger species at landfills, relative to their observed frequency in county-level species pools. Alternatively, if landfills also support a diverse community of native species, we would expect similar patterns of species richness, community composition and turnover between landfills and reference sites, as well as similar representation of dietary guilds at landfills and in county-level species pools.

## Materials and methods

### Community science data and study site selection

The eBird database, managed by the Cornell Lab of Ornithology, is the largest CS database of bird observations worldwide, and allows birdwatchers to document the species richness, and species abundance of birds seen at a given location, on a given date [27]. Summary data from over 100 million annual bird observations, vetted by volunteer reviewers to query unusual observations, are freely available to the public [27]. eBird data have been used to study species range expansion [28], to model population changes of migratory species [29], and to explore phenological shifts in migration [30].

In this study we collected eBird data on the encounter frequency of species from frequently-visited landfills across the US, nearby similarly sampled reference sites in the same county, and county species pools. eBird summarizes checklist data and publishes them as histograms, which show the percentage of complete checklists (i.e., those for which observers

indicated that they recorded all the birds they saw) that each species appears on for a given time-step. Species recorded on incomplete checklists have their encounter frequency down-weighted (from 1 sighting to 0.0015 of a sighting) so that they do not overly influence their relative abundance in the dataset, while still being represented as having been seen. Multiannual data is binned into 48 time-steps (four per month) spanning the calendar year. Although checklist-level data including the original observer's species level-counts are available upon request, we chose to work with encounter frequency data [31] to avoid biases related to inaccurate counts and incomplete checklists.

We identified landfills by searching for the word "landfill" in eBird's "explore hotspots" feature (a hotspot is defined as a publicly viewable site that is visited by many birders). Assuming that landfills with a larger number of submitted checklists would more accurately reflect the true species richness of a site, we excluded landfills with less than 100 total checklists [32]. While many landfill sites have both active sites and covered portions, we wanted to focus this study on landfills with at least one active site, so we also excluded sites labeled "(covered)" on eBird. To maintain a representative sample of species across the annual cycle and include migratory species, we further excluded sites at which data was missing for more than 15 time-steps across the calendar year, and at which data was missing for more than three consecutive time steps. Additionally, we excluded sites labeled "restricted-access" to avoid sampling bias. These criteria, resulted in a total of 19 landfills being included in the analysis.

In order to do a pairwise comparison, reference sites were chosen within the same counties as each landfill by selecting the hotspot with the most similar number of total checklists submitted. Reference sites had to include at least 100 checklists and meet the additional criteria for public access and data across seasons. We excluded large hotspots that were amalgamations comprised of many smaller sites. For example, South Padre Island consists of 22 specific hotspots and one broad hotspot ("South Padre Island (LTC 034)") that represents the entire island. In this case, "South Padre Island (LTC 034)" would be excluded. Because eBird hotspot boundaries, and how much area of each site is accessible to observers, are not clearly defined, we did not include any criteria relating to habitat types in the selection of reference sites. However, the sites were comprised of nature trails, forests, reservations, and nature preserves and thus contained a variety of "natural" habitat types for each region (S1 Table). Additionally, we use the 2016 National Land Cover Database to present bird-relevant land cover classes within a four-kilometer radius from the center of each landfill and reference site (S2 Table). To compare landfill bird communities to the surrounding species pool, we also downloaded county-level data on the seasonal encounter frequency of species. Above the checklist level, the smallest spatial scale at which eBird data is aggregated is that of county-level. We chose to analyze at the county level rather than at the state level (the next finest scale in US eBird data), because states often comprise multiple ecoregions. While a given species may be common in one part of a state, it may have never been reported in another, so the county-level species pool is a more representative sample of possible species at a given landfill than the state-level. Analyses included all data up to December 2019. Before analysis the data was cleaned to remove hybrid or non-species level taxa.

We assigned dietary trait data to each species using EltonTraits, a database that contains species-level attributes of 9993 extant bird species [33]. Each species was assigned to one of five diet classes: "Omnivore," "VertFishScav," "Invertebrate," "PlantSeed," and "FruiNect." "VertFishScav" species are those that feed on vertebrates, fish, and carrion [28]. Omnivore species are defined as those that consume food from multiple categories, with no category making up 50% or more of their diet [33]. Populations that were recently split into separate species were assigned the diet class of their previous species name. For example, the Sage Sparrow, *Amphispiza belli*, was recently separated into two distinct species, the Sagebrush Sparrow,

*Artemisiospiza nevadensis*, and the Bell's Sparrow, *Artemisiospiza belli*, and as a result, both species were assigned the diet class given to the Sage Sparrow in EltonTraits.

We first conducted our analyses using the full list of species recorded at each site, and secondly with a restricted list of the most commonly encountered species, which we defined as species with an encounter frequency greater than 5%. The first analysis ensures that migratory and rare or hard-to-detect species are represented in the dataset, whereas the second ensures exclusion of species recorded by chance (e.g., flyovers) that are not 'using' the site. We present results using the first method (i.e., all species included), and only discuss the second (common species only) when patterns differed between the two.

Using the hotspot and county names provided in S1 Table, all data can be downloaded from the eBird website (ebird.org). Navigate to the "Explore" tab, click "Explore Hotspots," and search for each hotspot. Alternatively, search for each county in the "Explore Regions" search bar on the "Explore" tab. Once at an individual hotspot's (or county's) page, click "Bar Chart" under the "Explore. . ." section. At the bottom of the bar chart page, the "Download Histogram Data" button will initiate the download of a text file containing all data.

## Data analysis

All analyses were performed in the R programming language [34].

To test our first prediction that species richness at landfills differed from reference sites, we calculated the total species richness at each landfill and its paired reference site. Because these are count (discrete) data, the standard regression assumption of normality of residuals cannot be met. Therefore, we performed a paired Wilcoxon rank test for differences in species richness between our site types. We also assessed if there were differences in the communities found at landfills and at reference sites, by calculating site-specific species-level encounter frequencies (i.e., the number of checklists for a given site in which a species was recorded, divided by the number of complete checklists submitted for that site). We treated landfills and reference sites as separate communities, and performed a non-metric multidimensional scaling (NMDS) analysis, using ANOSIM to test for differences, in the vegan package [35]. This analysis used the encounter frequency for each species to calculate pairwise dissimilarities for each landfill and reference site within a shared county.

To test our second prediction that landfills share more species across space than do reference sites, we calculated pairwise species similarity between pairs of landfills and pairs of reference sites, using Jaccard's Index (JI):

$$JI = \frac{Shared\ Species}{Shared\ Species + Unshared\ Species}$$

We calculated JI for all pairwise combinations of landfills (and reference sites) and plotted the relationship between JI and inter-site distance [7]. If landfills are dominated by widespread generalist species, we expected to see higher values of JI for landfill pairs than for reference site pairs, and for the slope of the relationship between JI and inter-site distance to be shallower for landfills than for reference sites (i.e., lower species turnover across increasing geographic distances).

To test our third prediction that species feeding at higher trophic levels are overrepresented at landfills when compared to county-level species pools, we calculated the prevalence of each dietary group by summing the site-level encounter frequencies for all species within the 5 dietary categories defined in EltonTraits. We calculated the same quantities using the county-level data, and assessed differences in the frequencies of each dietary type using a chi-squared test. Finally, to assess any taxonomic-based differences in community composition, we

conducted an indicator species analysis using the indicspecies package [36] in R. This identified species that occurred more frequently at landfills than expected, relative to paired reference sites.

## Results

### Summary data and selected sites

We identified 19 landfills and 19 paired reference sites that met our criteria (Fig 2). The dataset included 7,277 checklists for landfills and 7,165 from paired reference sites; the number of checklists per site varied from 103 (Fountain Avenue Landfill in Kings County, New York) to 1225 (Cameron Co. landfill, Texas). A paired Wilcoxon test confirmed that landfills and reference sites were not differentially sampled (P = 0.359, V = 107). On average, landfills had 383 (min = 103, max = 1225) checklists submitted, while reference sites had 377 (min = 104, max = 1224). A complete list of landfills, reference sites, and the number of checklists per site can be found in S1 Table. The mean distance between paired landfill and reference sites was 18.2 km (min = 1.4 km, max = 40.0 km). The counties included in the analysis had an average area of 2708.8 km$^2$ (min = 250.8 km$^2$, max = 13569.0 km$^2$)

### Species richness, community composition, and turnover

Landfills had a median species richness of 144 species (min: 84, max: 214), while their paired reference sites had a median of 160 (min: 82, max: 225) (Fig 3A). A paired Wilcoxon test revealed a small but significant difference between these two medians (P = 0.023, V = 38.0). An ANOSIM test revealed a significant difference in community structure between landfills and reference sites (P = 0.0016) (Fig 3B). However, the magnitude of this difference is small (R = 0.143).

Species similarity, measured by JI, showed a decreasing but saturating relationship with inter-site distance for both paired landfill sites and reference sites (Fig 3C) For landfills, the overall median JI was 0.405 (min: 0.1418, max: 0.6991) while the same metric was 0.400 (min: 0.1516, max: 0.8401) for reference sites. Our 19 sites generated 171 pairwise comparisons for

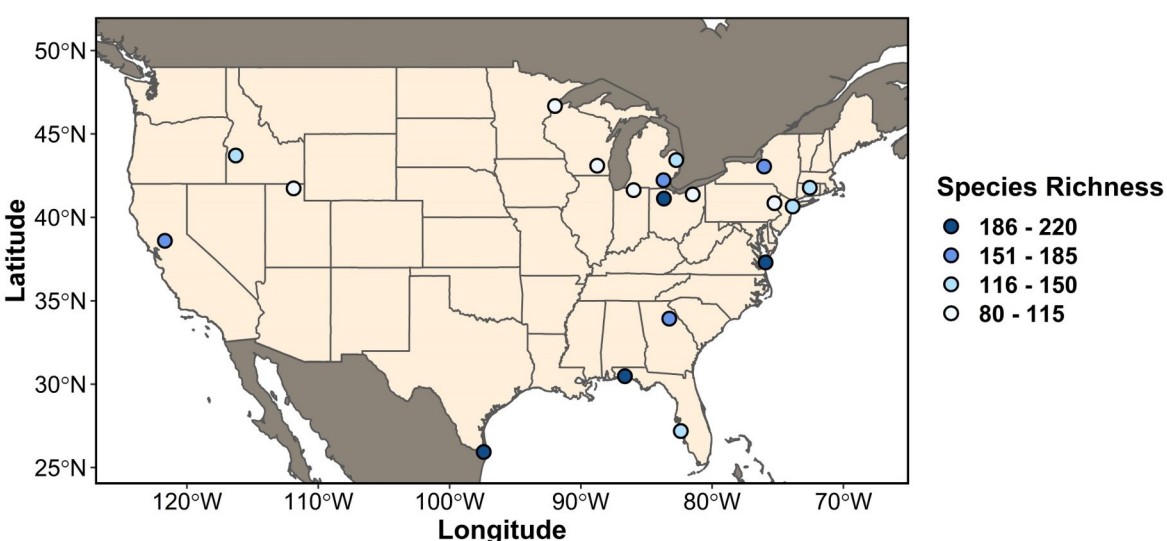

**Fig 2. Map of sites selected for analysis** [37]. The geographic distribution of landfills included in the analysis. Dot color represents the total number of bird species recorded by birdwatchers in the eBird database for each landfill.

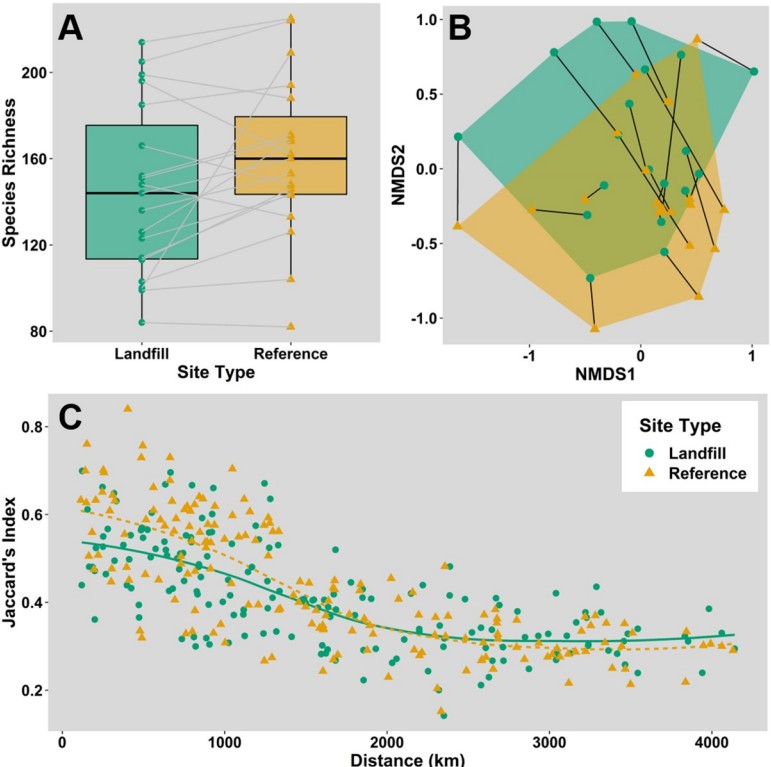

**Fig 3. Comparison of species richness, community composition, and species turnover.** Species richness and community comparison for landfills (green circles) and reference sites (orange triangles). Paired sites are connected by lines. (A) Box plot showing the distributions of species richness at selected landfills and reference sites. Horizontal lines and shaded box limits represent the median and interquartile range of species richness. (B) NMDS plot comparing communities, using species encounter frequencies. Using Jaccard's distance, 20 runs, and three dimensions, an acceptable stress value of 0.0773 was achieved. (C) Changes in community similarity (measured by Jaccard's Index) between pairs of landfills and pairs reference sites, plotted as a function of inter-site distances. Lines represent a spline fit.

each site type; due to lack of independence among these pairwise comparisons, we did not conduct formal statistical analyses. However, inspection of the plots of the relationship between JI and distance suggest no difference between landfill pairs than between reference site pairs. We also repeated this analysis using Sorensen's Index as a metric of community similarity; patterns of community similarity across space were similar to those using JI (S1 Fig).

**Indicator species analysis and dietary diversity.** The indicator species analysis revealed that thirteen species were encountered significantly ($P < 0.05$) more frequently at landfills than reference sites (Table 1). Of these, six species were gulls, one (European Starling, *Sturnus vulgaris*) is a widespread invasive species, three are migratory waterfowl and three are grassland birds. Notably, Ross's Goose (*Chen rossi*) and Slaty-backed Gull (*Larus schistisagus*) were seen very infrequently at landfills (respective encounter frequencies of 0.44% and 0.50%). However, they were seen even less at reference sites, resulting in significant differences (respective encounter frequencies of 0.13% and 0% respectively).

The relative frequency of dietary types showed no significant difference between landfills and their representative county-level species pools when all species were included ($P = 0.121$, $X^2 = 7.303$) (S2 Fig). However, when the analysis was limited to the most common species (i.e., those appearing on at least 5% of checklists), the diet classes at landfills did not represent a random sub-set of the county-level species pool ($P = 0.049$, $X^2 = 9.52$) (Fig 4). This difference is

**Table 1. Indicator species analysis results.**

| Species | Habitat Specialization | Encounter Frequency (%) | | R Statistic | P value |
|---|---|---|---|---|---|
| | | Landfill | Reference Site | | |
| Herring Gull (*Larus argentatus*) | Generalist | 44.44 | 14.33 | 0.527 | 0.0010 *** |
| Ring-billed Gull (*Larus delawarensis*) | Generalist | 46.47 | 22.32 | 0.499 | 0.0022 ** |
| Iceland Gull (*Larus glaucoides*) | Generalist | 12.49 | 2.24 | 0.438 | 0.0003 *** |
| Lesser Black-backed Gull (*Larus fuscus*) | Generalist | 10.30 | 0.95 | 0.415 | 0.0011 ** |
| European Starling (*Sturnus vulgaris*)† | Generalist | 53.50 | 34.51 | 0.410 | 0.0120 * |
| Glaucous Gull (*Larus hyperboreus*) | Generalist | 10.05 | 0.46 | 0.391 | 0.0009 *** |
| Savannah Sparrow (*Passerculus sandwichensis*) | Grassland | 12.46 | 5.85 | 0.372 | 0.0206 * |
| Ross's Goose (*Chen rossi*) | Wetland;Grassland | 0.44 | 0.13 | 0.360 | 0.0212 * |
| Eastern Meadowlark (*Sturnella magna*) | Grassland | 13.53 | 3.25 | 0.348 | 0.0145 * |
| Northern Shoveler (*Spatula clypeata*) | Wetland | 8.60 | 4.41 | 0.339 | 0.0120* |
| American Pipit (*Anthus rubescens*) | Grassland | 4.13 | 2.14 | 0.320 | 0.0032 ** |
| Ruddy Duck (*Oxyura jamaicensis*) | Wetland | 8.37 | 3.22 | 0.319 | 0.0234 * |
| Slaty-backed Gull (*Larus schistisagus*) | Generalist | 0.50 | 0.00 | 0.223 | 0.0439 * |

List of species which are encountered significantly more frequently at landfills than at reference sites. The R statistic measures the magnitude of the difference between encounter frequency at landfills and reference sites.

†widespread invasive species.

driven by the overrepresentation of carnivores (i.e., the "VertFishScav" diet class), accounting for 25.4% of species encountered at landfills, compared to 15.3% of the county-level species pool, and the underrepresentation of granivores ("PlantSeed" diet class), which represented

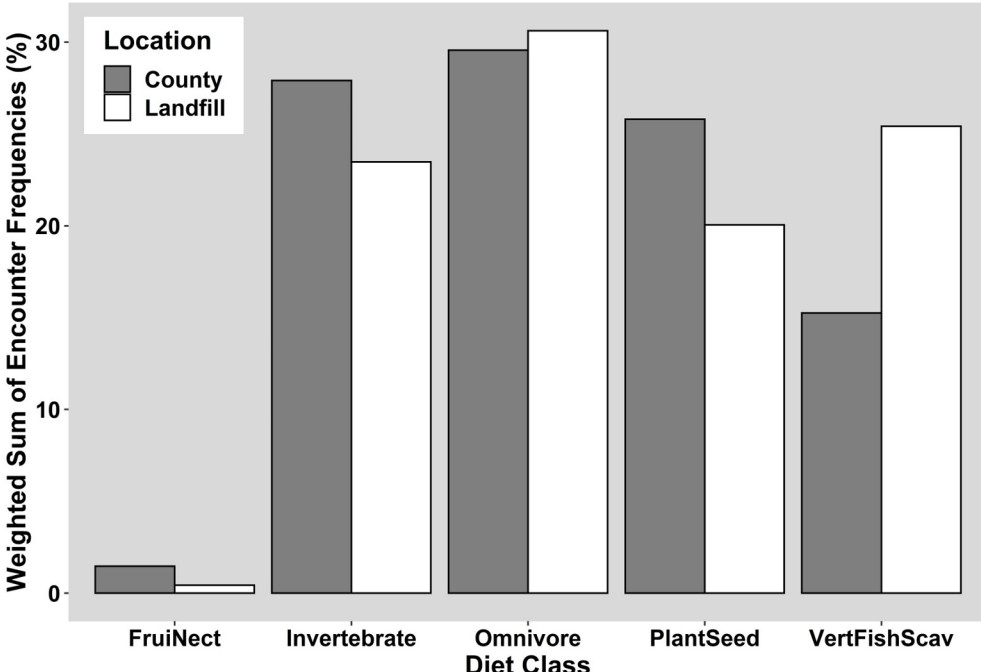

**Fig 4. Comparison of the common species' encounter frequencies for EltonTraits diet classes.** Frequency of the most common species' diet classes at landfills (white) compared to background county-level species pools (grey). All species with an average encounter frequency greater than 5% were included. For a detailed description of what each diet class means, see reference 33.

20.0% of species at the landfill level compared to 25.8% for the county. Additionally, there were more insectivores ("Invertebrate" diet class) recorded at the county level, 27.9% of species, compared to 23.5% at the landfill level.

## Discussion

Active landfills are attractive to many species of birds, including those perceived to be pest species, but additional habitat types contained within landfills may attract a wide-range of other species. Our goal was to quantify avian diversity at landfills relative to nearby reference sites and their county-level species pools. We predicted that if landfills were primarily attractive due to resource subsidies at active trash disposal sites, landfills would be less speciose, because they are dominated by dietary generalists and widespread human-adapted species that outcompete habitat specialists. However, we found only partial support for these predictions: median species richness was slightly lower (by 16 species) at landfills than reference sites, and there was no evidence for pronounced differences between species turnover at pairs of landfills and at pairs of reference sites. Although differences in community composition indicated that habitat generalists such as gull, are overrepresented at landfills relative to reference communities, our indicator species analysis also revealed that some habitat specialists (waterfowl and grassland birds) were also more likely to be encountered at landfills. Comparing landfills to county-level species pools showed that scavengers and carnivores are overrepresented at landfill sites, relative to background encounter frequencies. Together, these suggest that in spite of a signal of anthropogenic food waste at active sites attracting scavenging species, landfills may also have the potential to harbor comparable avian diversity to the surrounding landscape, and benefit some habitat specialists, even after refuse disposal sites are covered.

Matching past research, we found evidence that trash disposal sites attract scavenging and human-adapted dietary generalists, including gulls and invasive European Starlings (*S. vulgaris*) [15,38]. Because gulls and starlings use a variety of habitat types and foraging strategies, they benefit from readily accessible food resources in urban environments, including landfills [15,39,40]. As cavity nesters, European Starlings will nest in holes in walls, vents, as well as metal pipes [41], and thus likely use man-made structures at landfills for nesting.

We also found evidence that habitats contained within landfills, such as constructed wetlands, and grassland habitats, are attracting associated habitat specialists. While we did not perform any direct evaluation of habitat diversity of landfills compared to reference sites, our findings are consistent with the idea that habitat types found on landfill properties beyond active trash disposal sites contribute to bird diversity. Past studies have demonstrated a positive correlation between habitat diversity and bird species richness in human-modified urban parks [42]. Interestingly, two declining grassland species, Eastern Meadowlark (*S. magna*) and Savannah Sparrow (*P. sandwichensis*) were more likely to be encountered at landfills than nearby reference [43]. This is likely due to the creation and maintenance of grassland habitat on covered portions of landfills. Additionally, migratory waterfowl, including Ruddy Duck (*O. jamaicensis*) and Northern Shoveler (*S. clypeata*), were encountered more frequently at landfills than at reference sites. These birds are likely attracted to the constructed wetlands at landfills which are designed to limit widespread environmental impact of leachate [44].

These analyses highlight the potential for landfill sites to provide conservation value. With over 2,600 landfills in the US, each averaging 94 acres, there is a significant area of land dedicated to waste management [45]. After landfills are completed and covered, end uses are limited. However, completed landfills have been converted into a host of different assets including parks, hiking trails, and wildlife habitats [46,47]. Rahman and colleagues specifically suggest capped landfills could provide conservation value to grassland bird species [19]. Recent land-

use change has led to significant declines in suitable grassland habitats across the US [48]. Restored grasslands at landfill sites potentially offer opportunity for conservationists to work collaboratively with landfill managers. By planting native grasses, mowing outside of nesting season, and alternating mowing years, landfill managers can help provide habitat to species that have suffered drastic habitat losses and population declines. While concerns exist surrounding the bioaccumulation of toxic materials from constructed wetlands [49], there is also opportunity for properly designed wetlands to benefit migratory waterfowl populations [50].

This study provides one of the first multi-species analyses of landfill use by birds at a continental level. However, we note several limitations and biases related to site selection and CS data collection that should be considered in the interpretation of our results. First, mirroring past CS studies [51], the location of landfill sites meeting our minimum checklist number criterion was highly spatially biased towards areas of high human population density. Fifteen of our nineteen landfills were east of the Mississippi River, and the majority of those sites were located in the northeastern US. Second, due to a lack of information on the exact geographic size and habitat composition of eBird hotspots, landfill and reference sites were matched only by proximity and sampling efforts. Since geographic area is a known predictor of species richness [52], it is possible that our results could be biased if reference sites were systematically different in area to landfills. Similarly, reference sites were assumed to represent more "typical" wildlife habitats at the county level than landfills, although we were unable to test this explicitly. As the amount of data submitted to CS initiatives increase, future studies could expand this analysis to include additional landfills in underrepresented geographic regions, along with more standardized surveys of paired landfills and reference sites accounting for size and habitat types.

As with other CS studies, care must be taken to account for participant heterogeneities including identification skill, accuracy in counting and variation in effort. To minimize the effects of these heterogeneities in our analysis, we chose to work with summarized data that has been screened by trained volunteers to verify unusual observations. An alternative approach used in other studies is to work with the eBird raw dataset and filter observations according to eBird best practices, such as excluding incomplete checklists [53]. One advantage of using the filtered raw dataset is the possibility to work with abundance data, which would enable analyses not possible with species-level encounter frequency data (e.g., calculating proportional abundance of birds at higher taxonomic levels). However, abundance estimates from landfills could be heavily biased, with some species that aggregate at refuse (e.g., gulls) being too numerous to accurately count, while secretive or smaller-bodied species (e.g., sparrows) could be greatly undercounted. Further, summarized data includes occurrence data of species recorded only on incomplete checklists, but severely down-weights their encounter frequency, so that their presence is recorded without overrepresenting their encounter frequency.

In our analysis we chose to only include landfills with active sites, because these typically receive more visits from birdwatchers looking for species attracted to refuse, and also determine whether species of conservation interest are present while sites are still operational. We recommend that future research explore the differences between bird communities at landfills pre- and post-closure to determine whether habitat specialists increase their relative abundance once species attracted by trash are no longer present. Beyond birds, surveys of plant and animal taxa, are needed to quantify the biodiversity value of landfills more generally, and to inform conservation-based management practices at active and covered landfills. Given that many landfills have restricted areas not accessible to community scientists, targeted surveys carried out by researchers may be needed to more accurately document wildlife occurrence in these under-studied habitats.

In summary, our findings suggest the need for further study of bird communities at landfills in habitats other than active trash disposal sites. In particular, we recommend that future research should focus on the extensive grassland areas at covered landfills and associated wetland habitat that may have potential to support declining specialist species. Given that our analysis demonstrated the presence of threatened grassland birds at landfills across the US, with proper management landfill properties may present an opportunity for conservation.

## Supporting information

**S1 Fig. Comparison of species turnover between pairs of landfills and pairs of reference sites, using Sorensen's Index.** Changes in community similarity (measured by Sorensen's Index) between pairs of landfills and pairs reference sites, plotted as a function of inter-site distances. Lines represent a spline fit.
(TIF)

**S2 Fig. Comparison of the all species' encounter frequencies for EltonTraits diet classes.** Frequency of all species' diet classes at landfills (white) compared to background county-level species pools (grey).
(TIF)

**S1 Table. Complete list of sites included in the analysis.** American states are identified using two letter postal abbreviations. Checklist No. represents the total number of checklists submitted at each hotspot. Hotspot names appear exactly how they do in the eBird database.
(PDF)

**S2 Table. Percentage land cover of different bird-relevant land classes.** A buffer around the center of each site was chosen with a four-kilometer radius. The NLCD data uses a 16-class classification system [54], which we have reclassified into seven more bird relevant classes. NLCD classes 12 (Perennial Ice/Snow), 72 (Sedge/Herbaceous), 73 (Lichens), and 74 (Moss) did not appear in any of our sites. All of the developed landcover classes (classes 21–24) were grouped into one class labeled "Developed." The three forest classes (41–43) were aggregated into one class labeled "Forested." Classes 51–52, 71, and 81 were all counted as "Grassland" landcover. Classes 90 and 95 were grouped into the "Wetlands" class. The remaining classes (11, 31, and 82) were not aggregated with other land cover classes. We present the proportion of each of our reclassified land cover classes for all landfills and all reference sites.
(PDF)

## Acknowledgments

This project could not have been completed without the continued support of the Hall and Altizer labs of the University of Georgia. We would like to thank these labs for their support and feedback during this research.

## Author Contributions

**Conceptualization:** Zachary J. Arnold, Richard J. Hall.

**Data curation:** Zachary J. Arnold, Richard J. Hall.

**Formal analysis:** Zachary J. Arnold, Seth J. Wenger, Richard J. Hall.

**Investigation:** Zachary J. Arnold, Richard J. Hall.

**Methodology:** Zachary J. Arnold, Seth J. Wenger, Richard J. Hall.

**Project administration:** Zachary J. Arnold, Richard J. Hall.

**Resources:** Richard J. Hall.

**Software:** Zachary J. Arnold.

**Supervision:** Richard J. Hall.

**Validation:** Zachary J. Arnold, Richard J. Hall.

**Visualization:** Zachary J. Arnold, Seth J. Wenger, Richard J. Hall.

**Writing – original draft:** Zachary J. Arnold.

**Writing – review & editing:** Zachary J. Arnold, Seth J. Wenger, Richard J. Hall.

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
