## [Decision Letter · Decision Letter 0]

25 Feb 2021

PONE-D-20-34661

Not just trash birds: Quantifying avian diversity at landfills using community science data

PLOS ONE

Dear Dr. Arnold,

Thank you for submitting your manuscript to PLOS ONE. After careful consideration, we feel that it has merit but does not fully meet PLOS ONE’s publication criteria as it currently stands. Therefore, we invite you to submit a revised version of the manuscript that addresses the points raised during the review process.

We look forward to receiving your revised manuscript.

Kind regards,

Daniel de Paiva Silva, Ph.D.

Academic Editor

PLOS ONE

Additional Editor Comments:

Dear Arnold et al.,

After three independent reviews, I believe your manuscript may be suitable for publication in PLoS One after you improve your manuscript according to the reviewers' suggestions. All three reviewers raised important issues to be considered in your text. Still, pay special attention to the the suggestions raised by reviewer#2, who made and almost endless list of improvements. By the time you are about to resubmit your improved manuscript, please also provide a rebuttal letter containing all the improvements you made, considering the reviewers' suggestions. In case you do not agree with the proposed suggestion, please also explain why you believe such change is not adequate. Considering the current worldwide pandemic scenario, I believe a three months period will be enough for you to do all the required changes. In case you are able to do it before, do not hesitate to resubmit earlier. Therefore, please submit by May 26th 2021.

All the very best and please stay safe.

Daniel Silva

Journal Requirements:

2. We note that Figures 1 and 2 in your submission contain map/satellite images which may be copyrighted. All PLOS content is published under the Creative Commons Attribution License (CC BY 4.0), which means that the manuscript, images, and Supporting Information files will be freely available online, and any third party is permitted to access, download, copy, distribute, and use these materials in any way, even commercially, with proper attribution. For these reasons, we cannot publish previously copyrighted maps or satellite images created using proprietary data, such as Google software (Google Maps, Street View, and Earth). For more information, see our copyright guidelines: http://journals.plos.org/plosone/s/licenses-and-copyright.

2.1.    You may seek permission from the original copyright holder of Figures 1 and 2 to publish the content specifically under the CC BY 4.0 license. 

2.2.    If you are unable to obtain permission from the original copyright holder to publish these figures under the CC BY 4.0 license or if the copyright holder’s requirements are incompatible with the CC BY 4.0 license, please either i) remove the figure or ii) supply a replacement figure that complies with the CC BY 4.0 license. Please check copyright information on all replacement figures and update the figure caption with source information. If applicable, please specify in the figure caption text when a figure is similar but not identical to the original image and is therefore for illustrative purposes only.

Reviewers' comments:

Reviewer's Responses to Questions

**Comments to the Author**

1. Is the manuscript technically sound, and do the data support the conclusions?

Reviewer #1: Yes

Reviewer #2: Partly

Reviewer #3: Yes

2. Has the statistical analysis been performed appropriately and rigorously? 

Reviewer #1: Yes

Reviewer #2: Yes

Reviewer #3: Yes

3. Have the authors made all data underlying the findings in their manuscript fully available?

Reviewer #1: Yes

Reviewer #2: Yes

Reviewer #3: Yes

4. Is the manuscript presented in an intelligible fashion and written in standard English?

Reviewer #1: Yes

Reviewer #2: Yes

Reviewer #3: Yes

5. Review Comments to the Author

Reviewer #1: The authors used eBird data to compare bird communities between landfills and reference sites. They found that the landfills across the USA could provide benefits for some avian species with conservation concerns. This is an interesting paper and can provide implications for bird conservations at these landfills. I think the paper can be accepted while considering the following comments.

Lines 39-40, provide the name of two declining species

Line 133, provide details about the environmental conditions at these 19 landfills, e.g. in a supplementary table.

Line 134, how were the reference sites selected? What kind of environments were there? Habitats or environments in the reference sites could significantly determine the avian compositions.

In addition, there are few comparisons between studies on bird use of landfills or species compositions.

Reviewer #2: I think that the authors did a great job with this study. The objective is clear, the study was well design and the hypothesis to be tested are explicit. The authors made a very good coverage of relevant literature about the subject and make explicit and clear the research problem as well the relevance of the study for understanding how birds are responding to novel human land uses, in this case landfills and I found broadly relevant for the fields of urban ecology and land planning in urban areas. The manuscript is very well written, and the phrasing is clear. Paragraphs are well structured and logically connected. Methodology sections is fairly complete, and results are presented in a concise way. Discussion is well focused and developed. Overall, I consider the manuscript of great value. I congrats the authors for the hard effort invested in the preparation of the study. Nevertheless, I have a series of concerns about the manuscript, particularly the analytical part (statistical analysis) that I consider should be approached to make the study more robust and easily replicable in future studies. Also, below I point out to particular aspects of the manuscript per section.

TITLE:

I would skip: Not just trash birds. It sounds a little bit unnecessary this part in the title use trash birds.

INTRODUCTION:

General comments:

1.I think that the instead refers to wildlife in the introduction, author can use the term “birds”, because this paper is devoted to birds.

2.I believe that the transition to the research problem in the third paragraph is a little bit abrupt. I consider that although it provides the arguments concerned to habitat for threatened species and benefits for some particular bird clades (gulls), the justification for the work can include more applied arguments and not only their perceived benefits to some avian groups, what are the applications for management, restoration etc of knowing the diversity in landfills?

3.I would include in the introduction more studies which have compared the diversity of landfill and other land use types and not only include studies showing the positive and negative effects of landfills on bird diversity or particular species. It is important to make this, because it gives context in the introduction about the necessity to make these comparisons.

4. Another important point for the introduction is that important aspects of the predictions of the hypothesis are not completely developed in the introduction. Therefore, is not clear to the reader why should we expect lower turnover in pairwise comparisons of landfill sites compared to reference sites?. Why should we expect more generalist species? Or omnivorous species? Etc. I think that literature (which is extensive for urban bird communities) can be used in the introduction to highlight these points and guide the reader to the logic or the raison d'être of the hypothesis of this study.

Particular comments:

Line 51: I think that with only say biotic homogenization is enough for this sentence

Line 85: I would use only citizen science and exclude volunteer-led -science, because the term use thereafter is the abbreviature CS

Line 89: it says that the study assessed the patterns of trait diversity of birds, but trait diversity is usually understood as functional diversity and I did not see any metric of functional diversity analysed in the study?. Although diet is a trait, is uncommon in the literature refer to it as trait and ca be confusing. I would suggest changing the term and instead use: dietary guilds, because this was the aspect explored in the study.

Line 90: it is not necessary to say reported by birdwatchers, because this is already clear in the methodology that the data comes from ebird

Line 91: county-level species pools concept is suddenly introduced in the formulation of the hypothesis, but it could be great to explain in the introduction why is important to compare bird diversity of landfills with county- level species pools.

Line 92-93: I consider that in the hypothesis i should be specified what refence sites are the authors referring? because some reference sites (urban) can have similar or even lower levels of taxonomic diversity.

Line 93: It is not clear to me the expression: altered community composition? How can we explicitly define this in highly modified human landscapes? Because in these landscapes almost all bird communities have altered species composition? To make clearer I would more explicitly define in the introduction what are these reference sites.

Lines 93-94: To me higher species similarity and lower turnover are basically referring to the same, it sounds to me redundant. You can leave just one.

Line 95: Again, I think that the part of omnivorous and generalist diets is redundant, because omnivorous are generalist in diet, because they use different food items.

Line 96: It is not clear the term “background species pools”. This term is describing county-level species pools, reference sites or both?

Line 98: I consider that besides reference sites it could be added county-level species pools, because in the lines 90 and 91 it says that the study will compare landfill with these two categories.

Materials and methods

General comments:

1. I consider that the authors should more explicitly give the reasons for not analysing the raw data of the checklists for exploring the taxonomic diversity of the landfill bird assemblages. I think that the argument of use encounter frequency data to avoid biases is weak. First, because in the reference cited is not explained and explicit the way how to analyse bird community data, being more a study aimed to compare vocal activity of one species (see reference 26). Second, because there are several ways to approach these biases of ebird data. For example, to account for observer heterogeneity, authors can calculate observer expertise scores using mixed general additive models (see https://www.nature.com/articles/s41467-020-18230-0) for a recent example. Excluding checklists from observers with low experience can also reduce the bias. Third, use of raw data of the checklists allow for future studies to make comparisons of landfills with the present study. Particularly, for exploring temporal trends in taxonomic diversity. For these reasons, I strongly recommend to the authors to make the analysis of bird data using the original observed species level counts.

2.While I agree with the use of the Wilcoxon rant test for comparing the differences in species richness between landfills and reference sites because is count data, I wonder to know why the authors did not consider other statistical models to assess the differences between landfills and reference sites. For instance, generalized linear models using a poisson distribution can be used or even GAM models (see https://rss.onlinelibrary.wiley.com/doi/abs/10.1111/j.1467-9868.2010.00749.x). I consider that the authors should explore the possibility to use other statistical models and not only depended of a single statistical test to support their conclusions or at least provide arguments for not using other statistical models.

3. Although I agree with the use of the Jaccard index for doing the pairwise comparisons, I think that the other indices can be used as well. For instance, Sorensen index is proved to be useful for exploring community similarity using only incidence data such as Jaccard. However, Sorensen compared the number of shared species to the mean number of species in a single assemblage, giving a more local view than Jaccard which can greatly enrich the exploration of community similarity in the manuscript.

4. The authors mixed “reference sites” and “county species pools” in the formulation of the hypothesis and their predictions and is not clear whether both are synonyms or different.

Particular comments:

Lines 106-108: Besides these applications, has ebird data been used before to explore the diversity of landfills? if yes would be important to notice this here or perhaps better in the introduction.

Lines 109-111: It is not clear how similarly are sampled refence sites in the county to the landfills in terms of sample completeness? It would great if the authors describe in quantitative terms this similarity in sampling effort. For instance, do reference sites of >= 100 checklists were included?

Line 110: This line is a perfect place for defining what does it means county-level species pool. For people in the USA, this is clear but for readers of the rest of the world probably no.

Line 118-119: I sought reference 26 and I checked the material and methods of this publication, but I could not find an explicit and clear explanation about this critical point for the data analysis. I would request the author a better explanation for this data analysis procedure

Lines 125-126: Authors should explain in these lines why is important for sampling bird taxonomic diversity to only have landfills with active sites?. I’m curious about the reason.

Line 132: There is an additional bracket at LTC034

Lines 135-136: Here the authors explain better the criteria to select reference sites, but I believe that is a good idea to establish a threshold value of the reference sites to be compared with the landfills.

Line 138: does county level data exclude the data of the landfills?. I assume that yes, but perhaps this can be detailed more thoughtfully.

Line 142: Mention the year 2014 into the text is redundant because this is clear in the reference

Line 144: It is a good idea to separate the category “FruiNect.”, because it would be important to explore how frugivorous and nectarivores guilds are independently responding to lanfdills. Merge them in a single category can hide these responses. This separation is possible in the Whitman et al 2014 database

Lines 152-153: I recommend justifying with a reference the threshold value of 5% for the encounter frequency

Lines 167-171: It is not clear how site-specific species-level encounter frequencies are used in the non-metric multidimensional scaling (NMDS) analysis to explore the degree of overlapping between clusters. I agree with the use of NMDS and ANOSIN to tests this prediction, but I think that this point (how frequencies are used and visualized in the NMDS) should be explicitly described.

Lines 182-184: why should we expect a shallower relationship for landfills?. I recommend the authors explain this in the introduction to give context about these relationships.

Lines 186-189: why landfills are only compared with county level species pools?

Results:

General comments:

To me is not clear the results presented in the lines 241-251. Are these representative counties reference sites? or county level species pools? In the text is ambiguous. Figure says county level species pools. Authors should clarify this point. Furthermore, it is not clear why dietary guild comparison is not made with reference sites, if the core of the study is the comparison between landfills and reference sites.

Particular comments:

Line 194: Only say dataset is enough. It would omit “data” in this part. Otherwise sounds redundant.

Lines 212-214: It is not clear why there is pseudoreplication in the pairwise comparisons? In the lines 197 and 198, the authors say that the mean distance between pair landfills and reference sites was 18.2 km. Based in this mean I think that at least some points are enough spatially isolated from each other to avoid pseudoreplication or what is the nature of the pseudoreplication in this comparison. I’m lost in this part; I would like to hear an explanation of the authors.

Lines 214-215: Why there was no evidence that species similarity between landfill pairs was higher or decayed more slowly with inter-site distance than between reference site pairs? Which part of the results support this? Because there was a not formal statical test of this aspect.

Line 226: The next sentence: “geom_smooth 226 function and ‘loess’ method in ggplot” is unnecessary to include in the explanation of the figure 3.

Line 227: I consider reassess this title, because species richness is one facet (measure) of taxonomic diversity) and this already covered in the previous section and trait diversity see previous comment. In fact, this part looks more concerned to species composition.

Line 236: It could be a good idea in the table 1, specify the category column, is referring to the degree of habitat or diet specialization? Or both? Perhaps the column “reference” should be called “reference sites”?

Line 242: what are the representative counties? Reference sites or county level species pool?

Lines 248: Correct “granviores”

Discussion:

General comments

1. Is missing in the discussion the factors (ecological, human) that can drive the not divergence in species richness between landfills and reference sites. I suggest the author focus more in discuss these factors and much less in previous studies in landfills not directly related to the aim of the study.

2. Author should consider discussing more why some particular groups as gulls are benefitting of landfills compared to other urban human land uses? This group showed a positive response to landfills but is not discussed.

3. Foraging guilds such as nectarivores and frugivorous are not discussed, do these groups showed a particular prevalence to landfills? If not, why.

4. Some taxonomic groups are missed in the discussion. Besides gulls, waterfowl and some grassland species, particular passerine families (Passeriformes are the most abundant and diverse bird order) showed differences between landfills and reference sites?

5. How this study benefits from comparing landfills with county species pools? County species pools are missed in the discussion.

6. I missed a general conclusion and take-home message for this study. The author should clearly develop a conclusion at the end of the discussion and not only highlight knowledge gaps and future research work.

Particular comments:

Line 260: To me was not evident the comparison of landfills with county-level species pools. I consider that that the discussion mainly be focused to the comparison between landfills and reference sites because this what is observed in the results.

Line 261-262: I think that this point is dubious, because either landfills and reference sites are inside the urban matrix, and it not clear why resource subsidies reduce species richness.?

Line 263: Again, I would suggest using a strong statistical model to test this comparison.

Line 264-265: The phrasing of is part is not clear. I would rephrase it again and point out that species turnover was not pronounced in landfills compared to pairwise comparisons of refence sites. Because this is what is shown in the figure C.

Line 268-270: I think that the authors should be more cautious with this conclusion. First, because of the quality and nature of the ebird data, second, because of the temporal scale of the checklists surveyed. Third, because a null model expectation was not rule out, it not possible to ascertain whether these differences actually exists. Fourth, because of the geographical bias, many locations strongly concentrated in the north-eastern region of the country, the small difference detected by the statistical tests etc. I would leave as a possibility, “probably”.

Lines 273-277: from line 273 to 277 the information presented is not relevant for the discussion of the results of this study, I did not see any connection bird-aircraft collisions and increased disease risk with the aim of this study (compare taxonomic bird diversity of landfills and reference sites). I suggest omit this part and focus more in discuss a little bit more why gulls and starling are thriving in landfills?

Line 275.277: If there is not enough previous references to discuss the results , is better to avoid statements about the not utility of the literature and instead highlight knowledge gaps that can be explored by future research to understand better the value of landfills for conserving birds.

Lines 280-282: How the results of this study are consistent a positive correlation between habitat diversity and bird species richness in human-modified urban parks, when habitat diversity was not measure as the author claimed? Because habitat was not measured in this study, it hard to support that this study is proving evidence for the correlation between habitat heterogeneity and bird diversity

Line 282: Avoid starting a phrase with “of “

Lines 287-288: I think that this part: “These areas are regularly mowed to prevent the growth of deep-rooting species, which are believed to be able to pierce the containment system “is irrelevant for the present study

Line 289: Try to include also scientific names for the species mentioned in the text. Otherwise common names will not prove to be helpful when researchers search for information of particular species in searches using web of science, scopus or google scholar or directly into the papers and reviews etc.

Line 299: I ask: this part refer to suitable population declines of grassland birds? In that is the case, authors can make it more explicit

Line 307: I think that one of the main assets of this study is highlighted here. This study makes a continental scale. This should be more clearly highlighted in the introduction section.

Lines 318-319: See comments in the methodology about this point.

Line 321: I would not only say eBird but “citizen science initiatives” such as inaturalist, xenocanto etc.

Line 326-327: in the lines 301-303 this was already discussed, sounds a little bit repetitive

References

1. Avoid using hyperlinks to the cited references, see references: 7,44

2. Avoid capitalize the titles and names of the journals, see references: 14, 23,34 ,44

3. Avoid underlie, see references: 7,14,18, 30,43,44

Figures

Avoid white lines in the background of the figures

Include the description for the dietary categories used in the figure 4. For example, explain what it means VertFishScav etc.

Reviewer #3: This manuscript addresses an interesting question about bird community on landfills, these waste disposal areas normally are focus of studies investigating diseases risks with its associated bird species considered generalists and homogeneous than surroundings. The authors used a citizen science based data from eBird to compare richness and community similarity between landfills and reference sites. The methods were clear and well developed to standardize samples by checklists, stating limitations and biases related to site selection and clearly taking account during interpretation of results and conclusions. All conclusions were based on data analyzed with implications to direct management of landfills to improve its role in maintain habitat specific and threatened species. In general, the manuscript is well written, all data are publicly available, provide new insights about bird community of a few studied anthropogenic landscape, and an interesting title with high potential to be used on scientific communication texts. Based on characteristics stated above I recommend publish as it is.

6. PLOS authors have the option to publish the peer review history of their article (what does this mean?). If published, this will include your full peer review and any attached files.

Reviewer #1: No

Reviewer #2: **Yes: **Andres Angulo Rubiano

Reviewer #3: No

---

## [Author Response · Author response to Decision Letter 0]

9 Jun 2021

A document has been uploaded entitled "Response to Reviewers" that addresses each point raised by the reviewers. Additionally, a document entitled "Cover Letter" has been uploaded that summarizes our response to reviewers and addresses concerns from the editor about map copyrights

---

## [Decision Letter · Decision Letter 1]

16 Jul 2021

Not just trash birds: Quantifying avian diversity at landfills using community science data

PONE-D-20-34661R1

Dear Dr. Arnold,

We’re pleased to inform you that your manuscript has been judged scientifically suitable for publication and will be formally accepted for publication once it meets all outstanding technical requirements.

Kind regards,

Daniel de Paiva Silva, Ph.D.

Academic Editor

PLOS ONE

Additional Editor Comments (optional):

Dear Arnold et al,

Considering that all three reviewers decided for the acceptance of your MS, I am pleased to inform you that your MS was formally accepted for publication in PLoS One! Congratualations!

Sincerely,

Daniel Silva, Ph.D.

Reviewers' comments:

Reviewer's Responses to Questions

**Comments to the Author**

1. If the authors have adequately addressed your comments raised in a previous round of review and you feel that this manuscript is now acceptable for publication, you may indicate that here to bypass the “Comments to the Author” section, enter your conflict of interest statement in the “Confidential to Editor” section, and submit your "Accept" recommendation.

Reviewer #1: All comments have been addressed

Reviewer #2: All comments have been addressed

Reviewer #3: All comments have been addressed

2. Is the manuscript technically sound, and do the data support the conclusions?

Reviewer #1: Yes

Reviewer #2: Yes

Reviewer #3: (No Response)

3. Has the statistical analysis been performed appropriately and rigorously? 

Reviewer #1: Yes

Reviewer #2: Yes

Reviewer #3: (No Response)

4. Have the authors made all data underlying the findings in their manuscript fully available?

Reviewer #1: Yes

Reviewer #2: Yes

Reviewer #3: (No Response)

5. Is the manuscript presented in an intelligible fashion and written in standard English?

Reviewer #1: Yes

Reviewer #2: Yes

Reviewer #3: (No Response)

6. Review Comments to the Author

Reviewer #1: The authors have responded appropriately to all the comments I raised for the first edition. No more comments now.

Reviewer #2: I congrats the authors for approaching all my concerns in a very comprehensive way. The manuscript now has a stronger methodological basis and it clarifies some dubious points in the methods and results sections in the previous version. The authors made a great job in the methods and discussion sections improving and extending the manuscript to make it more valuable as an important study for understanding the responses of bird communities to urban ecosystems. I consider that the manuscript is suitable for publication.

Reviewer #3: (No Response)

7. PLOS authors have the option to publish the peer review history of their article (what does this mean?). If published, this will include your full peer review and any attached files.

Reviewer #1: No

Reviewer #2: **Yes: **Andres Angulo Rubiano

Reviewer #3: No

---

## [Editor Report · Acceptance letter]

17 Sep 2021

PONE-D-20-34661R1 

Not just trash birds: Quantifying avian diversity at landfills using community science data 

Dear Dr. Arnold:

I'm pleased to inform you that your manuscript has been deemed suitable for publication in PLOS ONE. Congratulations! Your manuscript is now with our production department. 

Kind regards, 

on behalf of

Dr. Daniel de Paiva Silva 

Academic Editor

PLOS ONE